# A Brief Review of Inherited Neuropathies: A Perspective from Saudi Arabia

**DOI:** 10.3390/brainsci15040403

**Published:** 2025-04-17

**Authors:** Ahmed K. Bamaga, Anas S. Alyazidi, Feryal K. Alali

**Affiliations:** 1Department of Pediatrics, Faculty of Medicine, King Abdulaziz University, Jeddah 21589, Saudi Arabia; abamaga@kau.edu.sa; 2Faculty of Medicine, King Abdulaziz University, Jeddah 21589, Saudi Arabia; feryal.k.alali@gmail.com

**Keywords:** neuropathy, hereditary neuropathy, hereditary motor and sensory neuropathies, Charcot–Marie–Tooth neuropathy, next-generation sequencing

## Abstract

Inherited neuropathies are a heterogeneous group of disorders that affect the peripheral nervous system, leading to motor, sensory, and autonomic dysfunction. These disorders are classified into various subgroups, including hereditary sensory and motor neuropathies, distal hereditary motor neuropathies, hereditary sensory and autonomic neuropathies, and more complex forms. Advances in genetic testing, particularly next-generation sequencing (NGS), have significantly improved the identification of these disorders. Emerging therapies, such as gene therapy, small molecule therapies, and antisense oligonucleotides, offer promising treatment options. However, current treatments remain limited, and their clinical benefits in humans are not yet fully established. This review provides a comprehensive overview of recent developments and evolving therapeutic options for hereditary neuropathies, focusing on gene therapy, small molecule therapies, and antisense oligonucleotides. It also highlights the current state of inherited neuropathies in Saudi Arabia, emphasizing the need for national guidelines, patient registries, and collaborative research efforts. By integrating advanced genomic technologies and fostering international collaboration, we can improve the diagnosis, management, and treatment outcomes for patients with inherited neuropathies.

## 1. Introduction

Hereditary neuropathies encompass a diverse group of genetically inherited disorders that affect the peripheral motor, sensory, and autonomic nerves [1]. These disorders are categorized into four major subgroups: hereditary sensory and motor neuropathies, distal hereditary motor neuropathies, hereditary sensory and autonomic neuropathies, and more complex hereditary neuropathies [2]. The advent of advanced genetic testing, particularly next-generation sequencing (NGS), has significantly enhanced the identification of these disorders [3,4]. These tests enable the detection of genetic mutations that compromise the integrity of either the axon or myelin of peripheral nerves, leading to the development of neuropathies [5].

In response to the increasing identification of hereditary neuropathies, there has been a corresponding surge in therapeutic options [6]. Among the most promising treatments is gene therapy, which has advanced due to improvements in gene manipulation technologies [7]. Gene therapy aims to correct or replace defective genes responsible for neuropathies, offering a potential long-term solution. Another emerging approach is the use of small molecule therapies (SMTs), which target specific pathways involved in the pathology of hereditary neuropathies [8]. SMTs modulate disease processes at the molecular level, providing targeted treatment options that may alter the disease course.

Antisense oligonucleotides (ASOs) represent another innovative therapeutic strategy. ASOs are short deoxyribonucleic acid (DNA) and ribonucleic acid (RNA) molecules designed to bind to specific mRNA sequences, thereby modulating gene expression [9,10]. ASOs are relatively easy to deliver, as they do not need to integrate into the genome and are simple to synthesize. They have shown potential in treating various genetic disorders, including hereditary neuropathies, by altering the expression of disease-causing genes [10].

Despite these advancements, symptomatic treatment remains essential for managing many hereditary neuropathies. Current treatment options are limited, and while several therapies have shown efficacy in animal studies, their clinical benefits in humans are not yet fully established [6]. This review explores recent updates and evolving therapeutic options for major inherited neuropathies, focusing on the latest advancements in gene therapy, small molecule therapies, and antisense oligonucleotides. Our goal is to provide an overview of the current state of treatment and future directions in managing hereditary peripheral neuropathies.

## 2. Review

### 2.1. Overview on Major Inherited Neuropathies

Inherited neuropathies affect approximately 1 in 5000 children, with variations in prevalence depending on the specific disorder [11]. These disorders are a diverse group of genetic conditions that impact the peripheral nervous system, which transmits signals between the central nervous system (the brain and spinal cord) and the rest of the body (Table 1). Clinical manifestations of these neuropathies can vary significantly, even among individuals with the same genetic mutation, complicating diagnosis and treatment [12]. Despite significant progress in understanding the genetic and molecular mechanisms underlying inherited neuropathies, effective treatments remain elusive [1,2,12].

Current management strategies are primarily supportive, aiming to relieve symptoms, improve mobility, and enhance the overall quality of life for affected individuals. Targeted therapies that address the underlying causes of these disorders are increasingly being researched, with promising avenues including gene therapy, molecular chaperones, and high-throughput drug screening [6].

### 2.2. Charcot-Marie-Tooth Disease (CMT)

CMT is the most prevalent inherited neuropathy, affecting approximately 1 in 2500 people. It was first described in the late 19th century by British neurologist Howard Henry Tooth and French neurologists Jean Martin Charcot and Pierre Marie [13]. CMT can be subdivided into demyelinating and axonal forms based on their pathogenesis. Axonal forms primarily involve axon degeneration, while demyelinating forms involve myelin sheath degeneration preceding axon damage [14].

Despite extensive research efforts, there are currently no effective pharmacological treatments available for patients with Charcot–Marie–Tooth disease (CMT). The development of effective therapies faces several significant challenges: (1) the extensive genetic heterogeneity of CMT, with over 1500 identified mutations, including the notable 1.4 Mb duplication in PMP22 associated with CMT1A, which leads to overlapping and variable disease phenotypes; (2) the rarity of individuals with specific genotypes, which limits both research interest and pharmaceutical investment; and (3) the inherent complexity of translating findings from preclinical studies in rodent and cellular models to successful human clinical trials [15].

Among the promising treatments explored, ascorbic acid (vitamin C) was one of the first therapies evaluated for CMT1A. Although widely studied, its efficacy remains unproven. Preclinical in vivo studies on C22 mice demonstrated that ascorbic acid reduced the expression of PMP22, a key protein implicated in CMT1A pathogenesis, and improved motor function [16,17]. Despite these encouraging preclinical results and the favorable safety profile of ascorbic acid, Phase III clinical trials in humans, which tested doses ranging from 1 to 4 g/day over two years, failed to demonstrate clinical efficacy [18].

Another investigational therapy, PXT3003, has shown promise in clinical trials for immune-mediated peripheral neuropathies, with significant improvements observed compared to placebo [8,19,20,21]. However, other therapeutic approaches, such as progesterone receptor antagonists and ACE-083, faced trial termination due to various issues, including lack of efficacy or safety concerns [19,22]. In contrast, therapies targeting P2X7 purinoreceptors and lipid supplementation have demonstrated satisfactory safety and tolerability in early trials, though their efficacy remains under investigation [19,23]. Looking ahead, a Phase IIa randomized, double-blind trial is set to evaluate the efficacy, safety, and tolerability of NMD670, administered twice daily for 21 days, in ambulatory adult patients with CMT1A.

In addition to pharmacological approaches, emerging technologies such as cellular reprogramming and high-throughput drug screening hold significant potential for advancing CMT research. Cellular reprogramming allows for the generation of patient-specific cell types, including stem cells, neurons, and glia, from somatic cells like fibroblasts or lymphocytes [24,25]. This technique enables the creation of disease-specific models for studying pathogenesis and screening potential therapies. However, challenges remain, such as developing subtype-specific differentiation protocols and co-culture systems to accurately model myelination and neuromuscular junction pathology in vitro [26,27] (Table 2).

### 2.3. Hereditary Neuropathy with Liability to Pressure Palsies (HNPP)

Hereditary neuropathy with liability to pressure palsies (HNPP) is a genetic disorder linked to deletions in the PMP22 gene, which contrast with PMP22 duplications that cause Charcot–Marie–Tooth disease type 1A (CMT1A). CMT1A is a progressive neuropathy characterized by muscle weakness, atrophy, sensory deficits, and reduced reflexes [28,29].

HNPP manifests clinically as painless recurrent pressure palsies that target specific nerves or nerve plexuses, including the brachial plexus, and lead to focal motor and sensory symptoms. Preventive management includes both symptom control and lifestyle adjustments to avoid nerve compression and injuries. Some treatments and interventions have the potential to worsen the medical condition. The use of vincristine in chemotherapy treatments has led to increased HNPP symptoms [30], and surgical procedures must be avoided because they can cause serious nerve trauma [31].

Healthcare providers currently emphasize preventive approaches, including nerve pressure avoidance and ergonomic adjustments in routine activities [32]. While there is no definitive cure for the condition, some treatment methods have demonstrated potential benefits. The therapeutic use of IVIG has shown symptom improvement during acute disease episodes. Liew and Lo’s 2017 case study demonstrated that IVIG treatment produced significant symptom relief in an HNPP patient [33], which aligns with previous research supporting IVIG as a potential immunomodulatory therapy [34] (Table 3).

### 2.4. Hereditary Sensory and Autonomic Neuropathies (HSAN)

Hereditary Sensory and Autonomic Neuropathies (HSAN) encompass a group of rare inherited disorders characterized by sensory dysfunction and varying degrees of autonomic dysfunction [35]. These disorders primarily affect the peripheral nervous system, leading to symptoms such as loss of pain and temperature sensation, ulcerations, and autonomic disturbances, including cardiovascular and gastrointestinal dysfunction. The classification of HSAN remains unresolved and contentious, with ongoing debates over the delineation of subtypes. Initially, Dyck and Ohta proposed a classification system based on four numerical subtypes (HSAN I-IV) [36]. However, subsequent discoveries of additional genetic mutations and clinical phenotypes have introduced complexity into the classification, leading to semantic controversies and the recognition of further subtypes [37].

Each HSAN subtype is genetically distinct, with mutations in specific genes implicated in the pathogenesis of the disorder. For example, HSAN I is often associated with mutations in the SPTLC1 gene, while HSAN III (also known as Familial Dysautonomia) is linked to mutations in the IKBKAP gene [38,39]. Understanding these genetic distinctions is critical for developing targeted therapies and improving diagnostic accuracy. Currently, management options for HSAN are largely supportive, focusing on alleviating symptoms and improving patients’ quality of life [40].

Recent research has shed light on the molecular mechanisms underlying HSAN1, a subtype caused by mutations in the SPTLC1 gene. Studies have identified the accumulation of two neurotoxic sphingolipids—deoxysphingolipids—in patients with HSAN1 [41]. This accumulation results from a mutant enzyme’s reduced affinity for its normal substrate, L-serine, leading to the production of toxic metabolites [42]. This discovery prompted a two-year randomized, double-blind, placebo-controlled trial to evaluate the effectiveness of L-serine supplementation in correcting biochemical and neurological abnormalities in HSAN1 patients. The trial demonstrated that L-serine supplementation significantly reduced levels of neurotoxic deoxysphingolipids, suggesting it as a promising first treatment option for HSAN1 [43] (Table 4).

### 2.5. Familial Amyloid Polyneuropathy (FAP)

Familial amyloid polyneuropathies (FAPs) are fatal multisystem conditions inherited as autosomal dominant traits that cause nerve damage through amyloid fibril deposits resulting from mutated transthyretin [44]. A particular mutation in the transthyretin gene leads to a hereditary polyneuropathy that emerges in adulthood and stands as the most severe form of its type [45]. According to the scientific literature, the management strategy for this disorder includes three essential steps [46]. The main strategy of disease-modifying targeted therapy aims to halt both production and accumulation of amyloid fibrils [47,48]. Management of this disorder involves liver transplantation (LT) because it replaces the mutant transthyretin (TTR) protein source or the application of transthyretin kinetic stabilizers such as tafamidis and diflunisal [49,50,51] (Table 5).

### 2.6. Refsum Disease

Refsum disease is a genetic disorder that produces peripheral neuropathy symptoms and occurs in both adult and infantile forms [52]. A deficiency in phytanoyl-CoA hydroxylase causes the condition because this enzyme is essential for metabolizing phytanic acid [53]. When phytanic acid breakdown is impaired by enzyme deficiency, it builds up in tissues and produces multiple clinical signs [54]. Refsum disease treatment relies on dietary changes and plasmapheresis as primary management approaches [55]. Patients must limit their consumption of foods rich in phytanic acid to manage their condition [54,55]. Although some symptoms improve with treatment, conditions such as retinitis pigmentosa and sensorineural hearing loss show lesser improvement [56,57,58,59].

### 2.7. Giant Axonal Neuropathy (GAN)

GAN is a rare inherited disorder impacting both the central and peripheral nervous systems, marked by the presence of abnormally large axons [60]. The clinical presentation of GAN can vary significantly, which often leads to the consideration of other conditions, such as classic infantile neuroaxonal dystrophy, Charcot–Marie–Tooth hereditary neuropathy type 4 (CMT4), and leukodystrophies, including arylsulfatase-A deficiency [61,62]. Despite this, the characteristics of GAN have been extensively documented in the medical literature [63,64,65,66].

Regarding treatment, a groundbreaking in-human trial began in 2015 to evaluate the efficacy of intrathecal administration of scAAV9/JeT-GAN, a gene therapy. This trial has shown promise in slowing motor decline in GAN patients but also highlighted several associated adverse events [67]. Detailing the adverse events in this phase 1 trial evaluating intrathecal scAAV9/JeT-GAN gene therapy for giant axonal neuropathy, safety analysis revealed 682 adverse events (AEs) over a median observation period of 68.7 months, with 129 (18.9%) deemed possibly treatment-related. Among 48 serious adverse events (SAEs), only one (fever with emesis) was linked to therapy. Common SAEs included scoliosis (9 participants), urinary tract infections (6), and upper respiratory infections (5). Two fatalities occurred due to disease progression, unrelated to treatment. Treatment-related AEs included cerebrospinal fluid (CSF) pleocytosis (13 participants), elevated CSF IgG index (13), leukocytosis (8), thrombocytosis (7), and headaches (7). Mild, transient hepatic transaminase elevations (grade 1–3) were observed in three participants. Immune responses, such as persistent AAV9 neutralizing antibodies and CSF pleocytosis, were frequent but asymptomatic. No dose-limiting toxicities or neuroinflammatory pathologies were identified, even at the highest dose (3.5 × 10^14^ vg) [67]. The safety profile of scAAV9/JeT-GAN underscores both the expected and novel challenges in intrathecal gene therapy. While most AEs reflected underlying disease progression or immunosuppressive regimens (e.g., glucocorticoid-related infections), immune activation—evidenced by CSF pleocytosis, elevated neutralizing antibodies, and T-cell responses to AAV9 capsid—highlight the inherent hurdles of AAV-mediated delivery. Persistent humoral immunity suggests potential barriers to redosing, a critical consideration for future trials. Notably, the absence of severe neuroinflammation or dose-dependent toxicity, despite widespread biodistribution, supports the feasibility of higher doses. However, the transient, manageable nature of hepatic and hematologic abnormalities suggests these are unlikely to preclude clinical use. The single SAE directly linked to therapy (fever) resolved rapidly, reinforcing the tolerability of the approach. These findings align with broader gene therapy safety trends, where immune responses dominate AE profiles, yet emphasize the need for optimized immunosuppression strategies to mitigate anticapsid immunity and enhance therapeutic durability [67]. Nonetheless, given the recent positive outcomes, ongoing studies are necessary to further assess the safety and efficacy of this gene transfer therapy (Table 6).

### 2.8. Congenital Hypomyelinating Neuropathy (CHN)

CHN represents a rare inherited peripheral neuropathy disorder [68]. Patients with this condition show nonprogressive muscle weakness together with diminished reflexes, low muscle tone, substantially reduced nerve conduction velocities, and incomplete development of nerve myelin sheaths [69]. There does not yet exist any targeted or definitive cure to treat this disorder. The treatment strategy utilizes symptom management together with complete supportive care to improve patients’ overall health and quality of life [70,71] (Table 7).

### 2.9. Tangier Disease

Tangier disease is a rare genetic condition where affected individuals exhibit very low levels of high-density lipoprotein in their blood serum [72]. Specific manifestations of Tangier disease dictate the treatment strategies that guide its management. Peripheral neuropathy represents an additional possible complication for patients with Tangier disease. While specific clinical treatments for this condition remain unproven, patients can benefit from supportive measures like transient bracing and personalized exercise programs [73] (Table 8).

## 3. Saudi Perspective and Existing Consensus on Inherited Neuropathies

Saudi Arabia exhibits one of the highest global incidences of inherited neuropathies, resulting in a substantial and growing population of affected individuals spanning pediatric and adult demographics [74]. These disorders, which are hereditary and predominantly affect the peripheral nervous system, are particularly prevalent in Saudi Arabia due to genetic factors and the high rate of consanguineous marriages [75,76]. Given the significant burden of these conditions, this review explores the evolving landscape of inherited neuropathies and their management within the Saudi context.

To address the complexity of these disorders, the Saudi Board of Pediatric Neurology curriculum emphasizes recognizing the pathological characteristics and clinical features of inherited neuropathies encountered in pediatric practice [77]. This educational framework ensures pediatric neurologists are equipped to diagnose and manage these conditions effectively. The curriculum covers a wide spectrum of inherited neuropathies, detailing their genetic basis, disease mechanisms, and clinical presentations. By integrating this knowledge into training, clinicians are better prepared to differentiate neuropathies, understand disease progression, and anticipate complications, improving diagnostic accuracy and enabling tailored treatments [78].

The medical community in Saudi Arabia acknowledges the challenges of diagnosing and managing inherited neuropathies, particularly late-onset forms in adults. Consensus has been reached on key issues: the rising incidence of these disorders [79], their genetic heterogeneity [79], and primary clinical features such as muscle weakness, sensory abnormalities, and impaired motor function [80]. A multidisciplinary approach and genetic counseling are increasingly emphasized as essential components of care [81,82].

Despite growing awareness, Saudi Arabia lacks national clinical guidelines for inherited neuropathies. Current management relies on broad evidence-based principles without standardized protocols [81]. This underscores the urgent need for stakeholders—neurologists, geneticists, researchers, and patient advocates—to collaborate on comprehensive national guidelines. A recent Spanish multidisciplinary initiative developing CMT guidelines serves as a model [82]. Standardized protocols would improve diagnostic criteria, treatment consistency, and data collection, driving innovation in care and research [82].

Inherited neuropathies pose significant challenges due to genetic diversity and clinical complexity [79]. Effective management requires comprehensive genetic and phenotypic data. Saudi Arabia’s lack of a national neuropathies registry highlights the need for a centralized repository to support research and clinical trials [80,81,82]. The Global Registry for Inherited Neuropathies (GRIN) offers a collaborative platform for data sharing [83].

A recent Saudi initiative systematically collects data on inherited neuropathy cases nationwide, aiming to identify novel genetic mutations and correlate them with clinical phenotypes. This effort will enhance understanding of disease mechanisms, improve diagnostic accuracy, and facilitate personalized therapies. By cataloging mutations unique to the Saudi population, this initiative lays the groundwork for targeted research and tailored treatments.

The rarity and genetic heterogeneity of these disorders necessitate data sharing. International databases like RD-CONNECT, DECIPHER, and GENESIS integrate NGS data with clinical phenotyping, providing invaluable resources for clinicians and researchers [84]. In Saudi Arabia, where inherited neuropathies affect up to six generations in single families, a national registry would offer insights into genetic risks, improve counseling, and connect families for shared learning [85].

Patient advocacy groups, including the Hereditary Neuropathy Foundation (HNF) and the American Association of Neuromuscular and Electrodiagnostic Medicine (AANEM), play a vital role in education and support. These organizations enhance knowledge dissemination and foster community networks critical for patient care.

In Figure 1, a summary of efforts and gaps in addressing inherited neuropathies in Saudi Arabia is presented, highlighting the need for continued investment in research, registry development, and multidisciplinary collaboration to improve patient outcomes.

## 4. Future Directions

While recent advances in disease-modifying therapies for inherited neuropathies—driven by insights into pathogenic mechanisms—offer transformative potential, future research must prioritize standardized, objective assessment tools to rigorously evaluate therapeutic efficacy, address heterogeneity in trial design, and ensure clinically meaningful outcomes across diverse genetic subtypes [12]. New studies should focus on developing gene therapy and precision medicine solutions targeting the specific genetic mutations responsible for CMT disease. Investigating the molecular mechanisms of CMT will help researchers identify new potential therapeutic targets. Establishing detailed patient registries and biobanks will enable longitudinal studies and the design of personalized treatments. Research efforts should expand to explore the pathophysiological mechanisms underlying HNPP. Advancements in genomic technologies, such as next-generation sequencing (NGS), will help identify additional genetic factors contributing to HNPP. Research should prioritize therapies targeting HNPP’s molecular pathways and establishing clinical guidelines to improve patient care.

Research priorities for hereditary neuropathies must align with the unique pathomechanisms and clinical challenges of each disorder. For HSAN, elucidating genetic drivers and biochemical pathways is foundational; such insights could inform targeted therapies, including neuroprotective agents or gene-editing approaches to correct mutations. Collaborative efforts to build international patient registries will strengthen genotype-phenotype correlations, enabling stratified therapeutic development. Similarly, in FAP, disease-modifying strategies should prioritize RNA interference or antisense oligonucleotides to suppress amyloidogenic transthyretin production, complemented by advances in biomarker discovery to facilitate early diagnosis and treatment monitoring. For Refsum disease, biomarker development is critical not only for early detection but also for tracking disease progression, which could optimize the timing of interventions such as gene therapy or enzyme replacement. In parallel, refining dietary protocols to restrict phytanic acid intake remains essential for mitigating acute exacerbations. In GAN, therapeutic development should center on gene therapies to restore gigaxonin function, supported by mechanistic studies exploring its role in neuronal integrity. Natural history studies and patient registries are indispensable for accurately assessing treatment efficacy and identifying novel therapeutic targets. Furthermore, CMT research must clarify molecular regulators of myelination to advance stem cell-based or myelin-repair therapies, while standardizing diagnostic criteria to reduce heterogeneity in clinical trials. For Tangier disease, therapies targeting cholesterol metabolism pathways—coupled with investigations into genetic and environmental modifiers—could enable personalized approaches. Global collaborative registries will enhance data quality across all these disorders, accelerating translational research and trial design. By integrating mechanistic discovery, therapeutic innovation, and collaborative data infrastructure, the field can address both shared and unique challenges in hereditary neuropathies.

Moreover, Saudi Arabia must establish a national inherited neuropathy registry to advance research and care. Developing specialized clinical guidelines requires collaborative networks of local and international experts. Successful management also demands public health initiatives focused on awareness and genetic counseling.

## 5. Conclusions

The landscape of inherited neuropathies is rapidly evolving, driven by advances in genetic research and therapeutics. The high prevalence of these conditions in Saudi Arabia underscores the need for focused research, comprehensive registries, and collaborative guideline development. By integrating advanced genomic technologies, robust patient support networks, and international partnerships, we can improve diagnosis, management, and outcomes. Personalized therapies hold significant promise for enhancing quality of life for affected individuals and families.

A dedicated multidisciplinary rehabilitation team is critical across nearly all disorders. Exploring therapeutic pathways specific to each condition remains essential. Establishing a national neuropathies registry in Saudi Arabia is not only beneficial but imperative. Such a registry would transform diagnosis, treatment, and research while advancing the country’s understanding of these disorders. By addressing challenges and leveraging global best practices, Saudi Arabia can build a system that benefits patients, families, and the medical community. Investing in this resource now will ensure better health outcomes for future generations.

## Figures and Tables

**Figure 1 brainsci-15-00403-f001:**
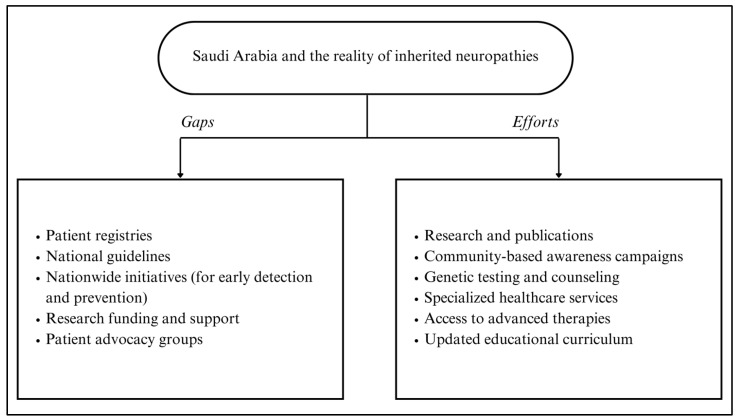
Summary of efforts and gaps to address inherited neuropathies in Saudi Arabia.

**Table 1 brainsci-15-00403-t001:** Classification, subtypes, and genetic involvement of inherited neuropathies.

Condition	Subtypes	Gene/Locus
Charcot–Marie–Tooth (CMT)	CMT1A	*Dup 17p (PMP22)*
	CMT1A	*PMP22 (point mutation)*
	CMT1B	*MPZ*
	CMT1C	*LITAF*
	CMT1D	*EGR2*
	CMT1E	*NEFL*
	CMT2A	*MFN2*
	CMT2B	*RAB7A*
	CMT2C	*TRPV4*
	CMT2D	*GARS*
	CMT2E	*NEFL*
	CMT2F	*HSPB1*
	CMT2G	*Unknown*
	CMT2H	*Unknown*
	CMT2I/J	*MPZ*
	CMT2K	*GDAP1*
	CMT4A	*GDAP1*
	CMT4B1	*MTMR2*
	CMT4B2	*SBF2*
	CMT4C	*SH3TC2*
	CMT4D	*NDRG1*
	CMT4E	*EGR2*
	CMT4F	*PRX*
	CMT4H	*FGD4*
	CMT4J	*FIG4*
	CMTX1	*GJB1*
	CMTX2	*Xp22.2*
	CMTX3	*Xq26*
	CMTX4	*AIFM1*
	CMTX5	*PRPS1*
	CMTX6	*PDK3*
Hereditary neuropathy with pressure palsies (HNPP)	HNPP	*Del 17p (PMP22)*
		*PMP22 (point mutation)*
Hereditary Sensory and Autonomic Neuropathies (HSAN)	HSAN IA	*SPTLC1*
	HSAN IB	*DNMT1*
	HSAN IC	*ATL1*
	HSAN ID	*SPTLC2*
	HSAN IE	*NGF*
	HSAN IIA	*WNK1*
	HSAN IIB: Caused by FAM134B gene mutations	*FAM134B*
	HSAN IIC	*KIF1A*
	HSAN IID	*SCN9A*
	HSAN III	*IKBKAP*
	HSAN IV	*NTRK1*
	HSAN V	*NGF*
		*NTRK1*
Familial Amyloid Polyneuropathy (FAP)	FAP Type I	*TTR (Val30Met)*
	FAP Type II	*TTR (Arg104His)*
Refsum disease	Classic Refsum Disease	*PHYH*
	Refsum Disease due to PEX7 Deficiency	*PEX7*
Giant axonal neuropathy (GAN)	GAN	*GAN*
Congenital hypomyelinating neuropathy (CHN)	CHN1	*MPZ*
	CHN2	*PMP22*
	CHN3	*EGR2*
	CHN4	*SOX10*
	CHN5	*CNTNAP1*
	CHN6	*KIF1B*
Tangier disease	Tangier disease	*ABCA1*

**Table 2 brainsci-15-00403-t002:** Recent studies on possible treatments for CMT disease.

Author	Year	Study Type	Intervention
Canals et al.	2023	Cross-sectional	Treatment of chronic pain in patients with CMT. The vast majority of patients reported effectiveness of cannabis in controlling their pain symptoms
Créange et al.	2023	Phase IIb	Treatment with high-dose pharmaceutical-grade biotin led to an improvement in various sensory and motor parameters
Bai et al.	2022	Preclinical	Treatment with IFB-088 improves neuropathy in CMT1A and CMT1B mice
Attarian et al.	2021	Phase III	A double-blind, placebo-controlled, randomized trial of PXT3003 for the treatment of CMT1A. PXT3003 doses were safe and well-tolerated
Gautier et al.	2021	Preclinical	AAV2/9-mediated silencing of PMP22 prevents the development of pathological features in a rat model of CMT1A

**Table 3 brainsci-15-00403-t003:** A recent study on a possible treatment for HNPP disease.

Author	Year	Study Type	Intervention
Vrinten et al.	2016	Double-blind, placebo-controlled n-of-one trial	A 35-year-old female patient was treated with IVIg (0.4 mg/kg/day) for 5 days followed by maintenance doses every 3 weeks, which led to improvements in muscle strength and resolution of pain.

**Table 4 brainsci-15-00403-t004:** Recent studies on possible treatments for HSAN disease.

Author	Year	Study Type	Intervention
Fridman et al.	2019	Randomized placebo-controlled trial	Patients aged 18–70 years with symptomatic HSAN1 were randomized to l-serine (400 mg/kg/day) or placebo for 1 year. All participants received l-serine during the second year.
Garofalo et al.	2011	Unspecified clinical trial	In mice bearing a transgene expressing the C133W *SPTLC1* mutant linked to HSAN1, a 10% l-serine–enriched diet reduced dSL levels. l-serine supplementation also improved measures of motor and sensory performance as well as measures of male fertility.

**Table 5 brainsci-15-00403-t005:** Recent studies on possible treatments for FAP.

Author	Year	Study Type	Intervention
Coelho et al.	2012	Randomized, double-blind trial	Patients received tafamidis 20 mg QD or placebo
Adams et al.	2000	Clinical trial	They underwent orthotopic cadaveric liver transplantation and received an ABO compatible graft, and received as immunosuppressive therapy a triple drug combination of cyclosporin A (Novartis, Basel, Switzerland), steroids and azathioprine (Gugenheim et al., 1987), or rabbit antithymocyte globulin (Merieux, France) in place of cyclosporin
Suhr et al.	2015	Clinical trial	Administration of patisiran led to rapid, dose-dependent, and durable knockdown of transthyretin, with the maximum effect seen with patisiran 0.3 mg/kg
Tokuda et al.	1998	Clinical trial	In an initial trial on three patients, TTR-adsorption therapy consistently reduced the serum concentrations of both total and Met30 TTR to about a half of the pre-adsorption levels
Pomfret et al.	1998	Clinical trial	Thirteen patients with FAP have undergone successful liver transplant surgery. Ten of 13 patients (77%) remain alive an average of 49 months (range, 17–64 months) after transplantation.

**Table 6 brainsci-15-00403-t006:** Recent studies on possible treatments for GAN.

Author	Year	Study Type	Intervention
Bharucha-Goebel et al.	2024	Dose escalation trial	The study conducted an intrathecal dose-escalation study of scAAV9/JeT-GAN (a self-complementary adeno-associated virus-based gene therapy containing the *GAN* transgene) in children with GAN.
Bailey et al.	2018	Clinical trial	IT delivery of AAV9/JeT-GAN in aged GAN KO mice preserved sciatic nerve ultrastructure, reduced neuronal IF accumulations and attenuated rotarod dysfunction.

**Table 7 brainsci-15-00403-t007:** A recent study on a possible treatment for CHN.

Author	Year	Study Type	Intervention
Belin et al.	2019	Genetic intervention study	The study crossed Q215X mice with a transgenic mouse overexpressing HA-Nrg1 type III in neurons (30). We observed an improvement in myelin thickness and NCV in Nrg1 type III: Q215X mice.

**Table 8 brainsci-15-00403-t008:** A recent study on a possible treatment for Tangier disease.

Author	Year	Study Type	Intervention
Oram	2005	Review	ABCA1-stimulating drugs have the potential to both mobilize cholesterol from atherosclerotic lesions and eliminate cholesterol from the body.

## Data Availability

No new data were created or analyzed in this study.

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
