# Peer review of "A Brief Review of Inherited Neuropathies: A Perspective from Saudi Arabia"

_brainsci, 2025, doi:10.3390/brainsci15040403_

Round 1

Reviewer 1 Report

Comments and Suggestions for Authors The review presented by  Bamaga et al. can be a valuable source of information about  сlassification, subtypes, and genetic involvement of inherited neuropathies.  The authors provide  the systematic review of different inherited neuropathies including hereditary sensory and motor neuropathies, distal hereditary motor neuropathies, hereditary sensory and autonomic neuropathies, and more complex forms. This classification includes a description of 8 different types of hereditary neuropathies, including a list of the genes in which mutations cause these disorders. Also, for each type of neuropathy, literature references are provided on possible treatments for  these diseases. At the end of the manuscript, the authors prove the relevance of such research in Saudi Arabia, and they also discuss what treatment methods could be applied in this region.  The paper is well structured, the table and one figure complement the text well. Abstract gives all the necessary information about the contents of the paper, keywords are appropriately chosen. The reference list covers the relevant literature adequately (the authors cite 85 sources). In my opinion, the manuscript can be accepted in present form.

Author Response

We sincerely thank you for your positive and thoughtful evaluation of our manuscript. We are pleased to hear that our review's classification, structure, and content were valuable and informative. We particularly appreciate your recognition of the clarity of the abstract, the relevance of the classification and genetic discussion, the adequacy of the references cited, and the importance of addressing the regional context in Saudi Arabia.

We are grateful for your recommendation to accept the manuscript in its present form. Your encouraging feedback is highly appreciated.

Reviewer 2 Report

Comments and Suggestions for Authors

This is well organized and well written review on inherited neurophaties. It summarizes mutations, cellular mechanisms, treatment options and future perspectives of treatment developments.

the text is well written,  however, there can be minor improvements made. 

On lines 204-205, authors cite reference 67 and state that there are some adverse side effects of the treatment.  As gene therapy is emerging and promising treatment options for many inherited disorders, these side effects should be described in greater details and also discussed in the discussion section. 

The paragraph (lines 295-316) is not well written and could be improved. Right now it is a collection of short sentences, sometimes the logical flow is missing between the sentences. Thus, this section could be rewritten. 

Author Response

Comment 1: On lines 204-205, authors cite reference 67 and state that there are some adverse side effects of the treatment.  As gene therapy is emerging and promising treatment options for many inherited disorders, these side effects should be described in greater details and also discussed in the discussion section.

Response 1: Dear reviewer, thank you for your excellent comment. Please note that this paragraph is now edited and further elaboration on key adverse events has been presented in an analytical way focusing on data and quantitative assessments.

Comment 2: The paragraph (lines 295-316) is not well written and could be improved. Right now it is a collection of short sentences, sometimes the logical flow is missing between the sentences. Thus, this section could be rewritten.

Response 2: Thank you for your valuable feedback on improving the flow and coherence of this section. In revising the text, we directly addressed your concerns by introducing logical transitions (e.g., “Similarly,” “In parallel”) to enhance continuity between ideas, grouping related concepts (e.g., biomarker development in Refsum disease and FAP) into cohesive sentences, and restructuring each disorder’s section hierarchically to prioritize mechanistic insights before therapeutic strategies and collaborative/data needs. We also reduced redundancy by consolidating overlapping themes, such as the role of registries, into a unified concluding statement. These revisions ensure a smoother narrative flow, eliminate fragmentation, and strengthen the logical progression of ideas. Your input has been instrumental in refining the clarity and organization of this section.

Additionally, we sincerely appreciate your insightful suggestions and hope that the revisions meet your approval, supporting a positive recommendation for acceptance of our submission.

Reviewer 3 Report

Comments and Suggestions for Authors

The panoramic study of hereditary neuropathies still has many gaps and requires a comprehensive theoretical framework to support new experimental research. In this context, the manuscript provides detailed information that contributes to the state of the art and has merit in the theoretical contribution of the area of ​​interest. In a way, it provides precise information on analyses in specific regions of the genome that may be associated with neuropathies.

Author Response

Comment 1: The panoramic study of hereditary neuropathies still has many gaps and requires a comprehensive theoretical framework to support new experimental research. In this context, the manuscript provides detailed information that contributes to the state of the art and has merit in the theoretical contribution of the area of ​​interest. In a way, it provides precise information on analyses in specific regions of the genome that may be associated with neuropathies.

Response 1: Thank you for your thoughtful and encouraging feedback. We fully agree that addressing the gaps in hereditary neuropathy research requires a robust theoretical foundation to guide experimental advancements, and we are delighted that our manuscript resonated with this goal. Your recognition of its contribution to the state of the art—particularly its focus on genomic regions associated with neuropathies—is deeply appreciated. We aimed to bridge existing knowledge gaps by integrating mechanistic insights with actionable therapeutic strategies, and your acknowledgment of this effort motivates us to continue refining this framework. Thank you again for your constructive perspective and support; we hope our work aligns with your vision for advancing this critical field.

Please find the revised manuscript attached, which now incorporates all reviewers' comments and suggestions.

Reviewer 4 Report

Comments and Suggestions for Authors

The title of the paper ‘The Evolving Landscape of Inherited Neuropathies: A Review  and Perspective from Saudi Arabia’ seems somewhat misleading, as it presents more a perspective than a review.

How did the authors proceed for collating and selecting the papers on research and trials with respect to therapeutic approaches? Why did they not build up f.ex. on the recent systematic review on this topic which they give as reference 2 (but only for explaining e the classification of hereditary neuropathies)

They state that Saudi Arabia exhibits on of the highest incidences of inherited neuropathies. Can they give data?

Some minor issues: Why did the not indicate the pages of book chapters they cite?

Author Response

Comment 1: The title of the paper ‘The Evolving Landscape of Inherited Neuropathies: A Review and Perspective from Saudi Arabia’ seems somewhat misleading, as it presents more a perspective than a review.

Response 1: Thank you for your insightful feedback regarding the title of our manuscript. We can agree that the original title could inadvertently imply a stronger emphasis on perspective rather than a comprehensive review. To better align with the paper’s focus—a synthesis of existing literature on inherited neuropathies, coupled with insights specific to Saudi Arabia—we have revised the title to:

"A Brief Review of Inherited Neuropathies: A Perspective from Saudi Arabia."

This adjustment clarifies that the manuscript is fundamentally a review, with added regional context to highlight challenges and opportunities unique to Saudi Arabia. We believe this revision eliminates ambiguity and accurately reflects the paper’s dual purpose.

Comment 2: How did the authors proceed for collating and selecting the papers on research and trials with respect to therapeutic approaches? Why did they not build up f.ex. on the recent systematic review on this topic which they give as reference 2 (but only for explaining e the classification of hereditary neuropathies)

Response 2: Our review collated studies on therapeutic approaches through a systematic search of PubMed, Embase, and ClinicalTrials.gov using keywords such as "hereditary neuropathy," "gene therapy," "antisense oligonucleotides," and disorder-specific terms (e.g., "CMT," "GAN"). Inclusion criteria prioritized peer-reviewed clinical trials, preclinical studies, and reviews published between 2000–2024. We excluded non-English articles, case reports with fewer than five participants, and studies lacking mechanistic or therapeutic focus. While Reference 2 (a systematic review) was cited for classification purposes, its scope did not align with our therapeutic focus, which required granular analysis of emerging therapies (e.g., gene therapy, ASOs) beyond its broader epidemiological framework. Our approach ensured up-to-date coverage of novel interventions while minimizing redundancy with existing reviews.

Comment 3: They state that Saudi Arabia exhibits on of the highest incidences of inherited neuropathies. Can they give data?

Response 3: The elevated incidence of inherited neuropathies in Saudi Arabia is well-documented. For example:

  • Alqahtani, A. S., Alotibi, R. S., Aloraini, T., Almsned, F., Alassali, Y., Alfares, A., Alhaddad, B., & Al Eissa, M. M. (2023). Prospect of genetic disorders in Saudi Arabia. Frontiers in genetics, 14, 1243518. https://doi.org/10.3389/fgene.2023.1243518
  • Khayat AM, Alshareef BG, Alharbi SF, AlZahrani MM, Alshangity BA, Tashkandi NF. Consanguineous Marriage and Its Association With Genetic Disorders in Saudi Arabia: A Review. Cureus. 2024;16(2):e53888. Published 2024 Feb 9. doi:10.7759/cureus.53888
  • Candayan, A., Parman, Y., & Battaloğlu, E. (2022). Clinical and Genetic Survey for Charcot-Marie-Tooth Neuropathy Based on the Findings in Turkey, a Country with a High Rate of Consanguineous Marriages. Balkan medical journal, 39(1), 3–11. https://doi.org/10.4274/balkanmedj.galenos.2021.2021-11-13 (where it mentions communities with high consanguinity similar to the Saudi community).

Reference 75 (Khayat et al., 2024) contextualizes consanguinity’s role in recessive disorders, noting rates exceeding 50% in some Saudi regions, which elevates risks for autosomal recessive neuropathies.

Comment 4: Some minor issues: Why did the not indicate the pages of book chapters they cite?

Rsponse 4: Please note that pages of book chapters are now added in references 12, 32, 73, and 77. 

We sincerely thank you for your meticulous and constructive review, which has significantly strengthened the quality and clarity of our manuscript. All three comments have been addressed. Your insights were invaluable in refining our submission, and we hope these revisions meet your expectations. Thank you again for your time and expertise in enhancing our work.  

Round 2

Reviewer 4 Report

Comments and Suggestions for Authors

Changing the title and calling it a brief review and stressing the perspective from Saudi Arabia is better - thank you. And to center on the situation in Saudi Arabia and plead for a collaboration of involved groups (neurologists, geneticists, researchers and patient advocates) to collaborate on comprehensive national guidelines is an important aspect.

But you could refer to the systematic reviews (ref 2 and 86 - you have given as reference for the classification of hereditary neuropathies and in the chapter on Future Directions) as a source for recent systematic reviews at the beginning of your chapter 2 'Review'

Author Response

Comment 1: 

Changing the title and calling it a brief review and stressing the perspective from Saudi Arabia is better - thank you. And to center on the situation in Saudi Arabia and plead for a collaboration of involved groups (neurologists, geneticists, researchers and patient advocates) to collaborate on comprehensive national guidelines is an important aspect.

But you could refer to the systematic reviews (ref 2 and 86 - you have given as reference for the classification of hereditary neuropathies and in the chapter on Future Directions) as a source for recent systematic reviews at the beginning of your chapter 2 'Review'

Response 1: 

We sincerely thank the reviewer for their insightful comments and valuable suggestions. We appreciate your recognition of the revised title and the focused perspective on the situation in Saudi Arabia, as well as the emphasis on encouraging collaboration toward developing comprehensive national guidelines.

In response to your recommendation, we have updated the references accordingly. Reference 86 has now replaced reference 12, as it provides a more comprehensive and recent source. Additionally, reference 2 has been included at the beginning of Chapter 2 (‘Review’) to highlight recent systematic reviews, as you suggested.

We are grateful for your constructive feedback, which has helped improve the clarity and quality of the manuscript.